# Laser Ablation Treatment of Recurrent Lymph Node Metastases from Papillary Thyroid Carcinoma

**DOI:** 10.3390/jcm10225295

**Published:** 2021-11-14

**Authors:** Chiara Offi, Claudia Misso, Giovanni Antonelli, Maria Grazia Esposito, Umberto Brancaccio, Stefano Spiezia

**Affiliations:** Department of Endocrine and Ultrasound-Guided Surgery, Ospedale del Mare, 80147 Naples, Italy; claudiamisso@hotmail.com (C.M.); giovanniantonelli81@libero.it (G.A.); mariagrazia.esposito@aslnapoli1centro.it (M.G.E.); umberto.brancaccio@aslnapoli1centro.it (U.B.); stefanospiezia@tiroide.org (S.S.)

**Keywords:** thyroid carcinoma, laser ablation, papillary thyroid carcinoma, lymph nodes metastases, recurrence

## Abstract

(1) Background: The incidence of papillary thyroid cancers is increasing. Papillary neoplasm metastasizes to the central and lateral lymph nodes of the neck. The recurrence rate is less than 30%. The gold standard of treatment for lymph node recurrences is surgery, but surgery is burdened by a high rate of complications. Therefore, laser ablation of recurrent lymph nodes has been recognized as an alternative treatment with minimal invasiveness, a low complication rate and a curative effect. (2) Methods: We analyzed 10 patients who underwent a total thyroidectomy and metabolic radiotherapy and who developed a lymph node recurrence in the laterocervical compartment in the following 12–18 months. (3) Results: Patients developed lymph node recurrence at IV and Vb levels in 70% and 30% of cases, respectively. All patients were treated with a single laser ablative session. Hydrodissection was performed in all patients. The energy delivered was 1120 ± 159.3 Joules and 3–4 Watts in 362 ± 45.7 s. No complications were reported. All patients underwent a 6-month follow-up. A volumetric reduction of 40.12 ± 2.2%, 49.1 ± 2.13% and 59.8 ± 3.05%, respectively at 1-, 3- and 6-months of follow-up was reported. (4) Conclusions: At 6 months, a fine needle aspiration was performed, which was negative for malignant cells and negative for a dosage of Thyroglobulin in eluate. The laser ablation is an effective alternative to surgical treatment.

## 1. Introduction

In the last decade, we have seen an exponential increase in the incidence of microcarcinomas (PTMC) and papillary thyroid carcinoma (PTC) due to easier access to imaging methods. This is associated with an increase in the diagnosis of synchronous and metachronous lymph node metastases. This event has opened new clinical and therapeutic scenarios [1,2,3]. PTC spreads by the lymphatic route; in 30% of cases it can be associated with synchronous lymph node metastases and in 30% of cases with metachronous lymph node metastases [4]. The gold standard treatment of both synchronous and metachronous lymph node metastases is surgery, which involves an extensive lymphectomy of the laterocervical and/or central compartments or a lymphectomy of one or more compartments. Surgery is burdened by a higher rate of major complications in the case of reoperations and in patients treated with metabolic radiotherapy, greater than in the primary intervention [4,5]. In recent decades, due to complications, several minimally invasive techniques have been perfected such as percutaneous ethanol injection (PEI), percutaneous laser ablation (LA), microwave ablation and radiofrequency ablation (RFA) [6,7,8,9]. US-guided LA has played a prominent role due to the ability to perform ablations of small lesions safely [7].

In 1960 the first handcrafted laser with a ruby was made [10]. Since then, the laser technique has undergone considerable developments and it has been applied in various medical fields. In 1962 the ruby laser was used for the first time in medicine, retinal surgery and dentistry [11,12,13]. Today, there are various types of lasers. The LA is a type of minimally invasive procedure. The use of LA must always be supported by scientific evidence, due to the complications and the side effects deriving from incorrect use.

LA was first used in the treatment of thyroid disease in the 2000s. Today it is routinely used in the ablation of symptomatic benign thyroid nodules, recurrence, and lymph node metastases. LA is used in the treatment of metastatic lymph nodes lesions due to the possibility of inducing small volumes of well-defined and predictable tissues necrosis. This feature makes LA suitable for the ablative treatment of metachronous metastases, adjacent to delicate anatomical structures.

Our study analyzes retrospectively the first 10 cases that underwent LA of the metastatic lymph nodes after a total thyroidectomy and radiometabolic therapy for PTC. The treated lymph node metastases occurred in the 12–18 months following the radiometabolic treatment with I131. Recurrence was diagnosed by serum thyroglobulin (Tg) elevation, a positive ultrasound, a fine needle aspiration positive for malignant cells and positive Tg in the eluate.

## 2. Materials and Methods

We retrospectively collected data from 10 consecutive patients undergoing LA treatment of metachronous lymph node metastases at the Department of Endocrine and Ultrasound-Guided Surgery of the “Ospedale del Mare”, Naples, Italy. The patients enrolled in the study were referred to our department in the period between December 2017 and June 2018 for the surgical treatment and in the period between June 2019 and December 2020 for the LA treatment. All included patients underwent a total thyroidectomy and subsequent metabolic radiotherapy (RAI) for the diagnosis of PTC, according to the American Thyroid Association (ATA) guidelines [4]. During oncological follow-up, all patients had an increase in serum Tg levels > 20 μg/L. Patients referred for LA treatment had a high surgical risk of comorbidities and/or had refused further surgery or were ineligible for a second RAI treatment. All cases were evaluated in the multidisciplinary oncological group prior to treatment.

The Campania Centro ethics committee approved the protocol and all patients signed a written informed consent. All procedures were performed in accordance with the Helsinki Declaration.

We included patients over 18 years with a histological diagnosis of PTC in the 12–18 months prior to ablative treatment. All patients were treated with a total thyroidectomy and RAI. During the follow-up, all patients showed an increase in serum Tg (values > 20 μg/L) and ultrasound evidence of pathological lymph nodes in the laterocervical compartment. All patients underwent a fine needle aspiration cytology (FNAC) of the lesion, which showed the presence of malignant cells and Tg values in the eluate >5000 μg/L. The volume of the lesion was calculated during the ultrasound examination. A complete history and preoperative hematological evaluation were collected in an electronic database. We evaluated patients’ hematological tests: complete blood cells count, thyroid function and thyroid antibodies. The B-mode ultrasound examination was performed to evaluate the lymph node level, the lesion’s volume and the presence of further metastases. Clinical and demographic data (age, sex, volume of nodal metastasis, level of metastasis) were collected in an electronic database.

Patients followed an ultrasound follow-up program after laser ablative treatment, with a timing of 1-, 3- and 6-months. During the follow-up, we calculated the treated lesion’s volume in ml, the volumetric reduction in percentage compared to the pre-treatment volume and the serum Tg value. At the end of the 6-month follow-up, we performed a lesion’s FNAC. In all treated cases we obtained a negative cytology for malignant cells and absence of Tg in the eluate.

The ultrasound examination was conducted with a 7.5–12 MHz linear probe (MyLab™ ClassC and MyLab™ 9 Platform, Esaote Biomedica, Genova, Italy). The basal volume of the lymph node lesion was calculated using the software. Vascularity was studied by Color Doppler (CD) examination and slow flow analysis.

We analyzed the number of laser fibers used, the Watts, the Joules and the time expressed in seconds. We used a continuous wave multi-source laser system with a length of 1064 nm (EchoLaser ModiLite^TM^, Elesta SpA, Calenzano, Italy). The procedures were performed with ultrasound guidance and with a free hand technique. Currently, EchoLaser can now be used with a recently released new software (available with the ESI, EchoLaser Smart Interface) for a more precise and safer planning of the procedure. The laser generator, connected to the optical fiber, produces an illumination of the optical fiber which generates heat in tissues adjacent to the tip by. The temperature increase induces the denaturation of the proteins and irreversible cell damage. The energy produced decreases exponentially distancing from the tip of the optical fiber.

The patients underwent a single LA session. The procedure was performed by placing the patients in a supine position with a moderate hyperextension of the neck. We performed a pericapsular local anesthesia with 2% Lidocaine.

The applicators used were 21 Gauche (G) needles. The output power varied from 3 to 5 Watts (W) and in our population we used an output power of 3 W in nine cases and a power of 4 W in only one case, due to the lymph node’s volume. The fibers used were quartz optical fibers with a flat tip and a diameter of 300 microns. The applicators were inserted into the target lesion through guides applied to the probe with different angles of incidence depending on the pre-treatment planning. The procedure began with the insertion of the 21G introducer into the target lesion along its major axis. The treatment was performed with a prefixed power, while the lighting time of the optical fiber varied according to the volume to be treated. The 21G applicators allow atraumatic, precise and multiple positioning of the fibers. More optical fibers can be used, as in one of the cases presented, with a distance of at least 0.8 cm. The tip of the fiber must have a minimum safety distance of 5–10 mm from the anatomical structures of the neck. We performed a preliminary hydrodissection with a 5% glucose solution to ward off the lymph node metastases from the anatomical structures of the neck [14,15,16].

We performed the “pullback” technique in one case. The “pullback” technique can be used in the treatment of lesions with a major axis greater than 2 cm [7]. The technique allows the enlargement of the area to be ablated and the planning made at the beginning of the procedure to be respected. On the ultrasound, the ablated area appears hyperechoic due to the micro bubbles created by the evaporation of liquids. The hyperechoic area increases its size as the volume of the necrosis area increases. The treatment can be considered concluded when the volume of the hyperechoic area remains stable. The evaluation of the treatment’s effectiveness is performed through the Color Doppler (CD) analysis [14,15,16,17]. The real volume of the area subjected to ablation can be determined 72 h after the end of the treatment, as the cell damage becomes permanent [14].

At the end of the procedure the functionality of the vocal cords was evaluated with an ecolaryngoscopy.

All patients were discharged two hours after the procedure in the absence of any type of symptoms. There were no cases of major or minor complications.

Statistical analysis was performed with SPSS version 23 (SPSS©, Chicago, IL, USA). Continuous variables were described as mean, standard deviation (SD) and range, while categorical variables were described as number of cases and the percentage.

## 3. Results

We retrospectively enrolled 10 patients (5 males and 5 females) treated with a single LA session for a single lymph node recurrence. All patients had a diagnosis of classic variant PTC that had been previously treated with a total thyroidectomy and RAI. The demographic characteristics are showed in Table 1. The mean age was 40.2 years (±17.98 SD). All patients were treated with suppressive therapy with L-thyroxine. During the oncological follow-up, all patients had an increase in the serum Tg value greater than 20 μg/L. The increase in serum Tg was followed by an ultrasound of the neck which revealed the presence of disease. There were seven metastatic lymph nodes at level IV and three at level Vb with a baseline volume of 1.82 mL (±3.45 SD). In all cases, a FNAC of the suspected lesion was performed with a Tg assay on the eluate. Data are reported in Table 1 and Table 2.

The size and margins of the metastases determined the number of laser fibers used. In metastases of less than 1 cm of maximum diameter (nine cases), only one fiber was used, while in metastases greater than 1 cm of maximum diameter (only one case), two laser fibers were used.

The treatment protocol, W and Joules, was planned before the start of the procedure by the expert operator (S.S.). An output power of 3 W was used in nine cases, while a power of 4 W was used in only one case for a mean of 531.86 s (±109.5 SD) and a mean of 1256 Joules (±396 SD). The data relating to individual cases analyzed are shown in Table 2.

After laser ablation treatment, the mean volume of the nodule was 1.12 ± 2.18 mL with a 40.12% reduction in volume 1 month after treatment. After 3 months the mean volume was 0.88 ± 1.65 mL with a 49.1% reduction in volume and after 6 months the mean volume was 0.706 ± 1.30 mL with a 59.8% reduction in volume (Table 3). The data showed a volume reduction of 60% after 6 months, but the most significant data were the absence of malignant cells at the FNAC performed after 6 months, the absence of Tg in the eluate and absent serum Tg values.

## 4. Discussion

The treatment of PTC is a total thyroidectomy with or without central lymphadenectomy followed or not followed by RAI therapy depending on the histotype and the presence of mutations [4]. Lymph node recurrence can occur in about 1–2% of patients after surgical and RAI therapy [4]. Lymph node recurrences can be treated with surgery or with additional RAI therapy, even if reoperations are burdened by a very high rate of major complications (25% of cases of vocal cord palsy) and metabolic radiotherapy is burdened by a very high failure rate when repeated [4]. In fact, various studies have shown that neoplastic cells subjected to repeated RAI therapies lose the ability to pick up radioactive iodine [4]. Ultrasound-guided mini-interventional procedures are an effective and safe alternative [6,7,8,9]. The minimally invasive ultrasound-guided procedures have been developed and validated. They can be performed repeatedly without an increased complication rate, without hospitalization and general anesthesia; all characteristics well accepted by patients. PEI was the first treatment used but was soon abandoned due to the unpredictable spread of alcohol [18]. Subsequently, RFA took over due to its ability to generate a complete necrosis of the lesion, even if burdened by high complication rates [19]. LA appears to be the safest and the most effective technique.

The LA technique of body tissues was proposed in 1983 by Bown and has undergone numerous revisions, standardizations and experiments [20]. Numerous data have emerged from the literature in support of minimally invasive ablative techniques. Dupuy et al. [21] treated eight cases of lymph node metastases with RFA; all healed with a single application but two cases of major complications were reported. Kim et al. [22] reported a series of 73 patients with lymph node recurrence, 27 treated with RFA and 46 treated with surgery. The study did not show statistical differences between the two groups in relation to disease free survival at 1 and 3 years. Unlike other techniques, LA is preferred in the treatment of thyroid pathologies due to the size of the introducers, which are reduced, easily reaching the target tissue with a low risk of tissue injury and bleeding. The thermic energy used allows a precise ablation of small volumes, reducing heat dissipation and the subsequent necrosis of the surrounding tissues. In cases of large lesions, multiple needles can be used simultaneously to increase the volume of the necrosis area. Mauri et al. [3] presented a series of 24 patients treated with LA for PTC’s lymph node recurrence. The study showed that 86.9% of patients had a complete ablation 30 months after treatment and in 79.1% of patients no disease was detected on imaging methods. In only five cases a second session was required and no increase in the complications rate was reported. Zhou et al. [23] published a study on 81 patients with PTMC, 36 treated with LA and 45 treated with a lobectomy with a unilateral sixth compartment lymphectomy. The study showed that, with a mean follow-up of 49.2 months and 48.5 months, respectively, no patients developed lymph node recurrences. Therefore, LA can be considered an alternative treatment for patients who are unsuitable for or who refuse surgery.

Our series of 10 patients showed that a normalization of serum thyroglobulin levels, the absence of neoplastic cells in the control FNC and the absence of Tg in the eluate were recorded 6 months after the ablative treatment. In only one case we used two laser fibers due to the large size of the lesion and neither major nor minor complications were reported. The risk of ablative treatment in relation to lymph node recurrences is thermic injury to the nerve structures; we reduced this risk by practicing a hydrodissection, with a 5% glucose solution, of the lesion and reducing the power and increasing the application time. Figure 1 shows one of the cases included in our case series. Image A shows the lymph node recurrence prior to the treatment. Image B shows the tip of the laser fiber in the center of the lesion before the ablative treatment was started. Image C shows the ultrasound at 6 months. The lymph node lesion appears as a hypoechoic area with a hyperechoic halo (image of scarring fibrosis).

Our data demonstrate that LA of PTC’s lymph node recurrence is a viable treatment. This technique allows a reduction in the surgical treatment of metachronous metastases from PTC with the reduction in morbidity associated with a reoperation.

It is useful in cases of multiple lymph node recurrences occurring during the follow-up, even after previous treatment. All these patients would be candidates for a lymphadenectomy with an exponential increase in complications.

However, our study had limitations: the sparse case history, the absence of randomization of the population, the impossibility of comparison with other techniques and the lack of follow-up at 5 and 10 years. This is a preliminary study; further analyses are needed.

## 5. Conclusions

LA represents a safe, effective and minimally invasive alternative to surgery in the treatment of lymph node recurrences from papillary thyroid carcinoma treated with surgical therapy and metabolic radiotherapy. This technique allows a total necrosis of the metastatic lesion associated with a reduced complications rate compared to surgical treatment.

## Figures and Tables

**Figure 1 jcm-10-05295-f001:**
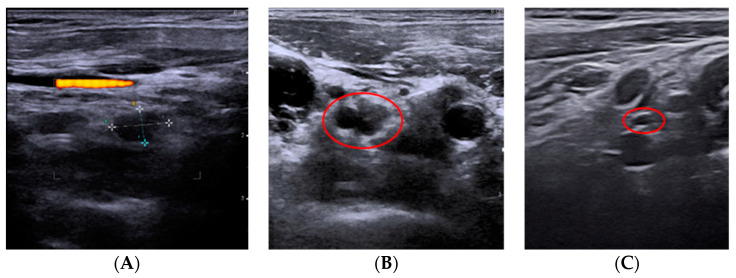
The images (**A**) show a lymph node recurrence to treat with laser ablation. Image (**B**) shows the tip of the laser fiber in the center of the lesion before the start of the ablative treatment. Image (**C**) shows the ultrasound follow-up at 6 months. The lymph node recurrence appears as a hypoechoic area with a hyperechoic halo (image of scarring fibrosis).

**Table 1 jcm-10-05295-t001:** Demographic and clinical data of the study population (FNAC, fine needle aspiration cytology; Tg, thyroglobulin; mL, milliliter).

Data	Population
Gender	
Male	5 (50%)
Female	5 (50%)
Age	40.2 ± 17.98
FNAC	Positive for malignant cell
Eluate Tg	>5000 μg/L
Lymph node level	
IV	7 (70%)
Vb	3 (30%)
Side	
Right	4 (40%)
Left	6 (60%)
Lymph node volume	1.82 ± 3.45
Fiber optic	1.01 ± 0.31
Watt	3.1 ± 0.31
Time in second	531.86 ± 109.5
Joule	1256 ± 396
1-month volume in ml	1.12 ± 2.18
% reduction at 1 month	40.12 ± 2.2
3-months volume in ml	0.88 ± 1.65
% reduction at 3 months	49.1 ± 2.13
6-months volume in ml	0.706 ± 1.30
% reduction at 6 months	59.8 ± 3.05

**Table 2 jcm-10-05295-t002:** Demographic and clinical data of the included patients.

Patient	Gender	Age	FNAC	Eluate Tg (μg/L)	Lymph Node Level	Side	Lymph Node Volume (mL)	Fiber Optic	Watts	Time (Seconds)	Joules
**1**	Male	20	Positive	>5000	IV	Right	0.7	1	3	270	810
**2**	Male	19	Positive	>5000	IV	Right	0.35	1	3	365	1095
**3**	Male	19	Positive	>5000	IV	Right	0.25	1	3	365	1095
**4**	Female	59	Positive	>5000	Vb	Left	11.58	2	4	340	2720
**5**	Female	59	Positive	>5000	IV	Left	0.46	1	3	335	1005
**6**	Female	63	Positive	>5000	IV	Left	1.27	1	3	420	1260
**7**	Male	28	Positive	>5000	IV	Right	1.32	1	3	400	1200
**8**	Female	35	Positive	>5000	VB	Right	1.05	1	3	395	1185
**9**	Female	48	Positive	>5000	VB	Left	0.79	1	3	325	975
**10**	Female	52	Positive	>5000	IV	Left	0.98	1	3	405	1215

**Table 3 jcm-10-05295-t003:** Clinical data of volume reduction in the lymph nodes metastases treated with laser ablation.

Patients	1 Month Volume (mL)	% Reduction at 1 Month	3 Months Volume (mL)	% Reduction at 3 Months	6 Months Volume (mL)	% Reduction at 6-Months
**1**	0.1	42	0.08	50	0.07	60
**2**	0.22	37.2	0.19	45	0.16	55
**3**	0.15	40	0.13	48	0.11	56
**4**	7.3	37	5.56	52	4.4	62
**5**	0.22	43	0.2	47	0.26	57
**6**	0.74	42	0.62	51	0.48	62
**7**	0.76	42	0.64	51	0.49	63
**8**	0.61	41	0.53	50	0.38	64
**9**	0.48	39	0.4	49	0.32	59
**10**	0.61	38	0.51	48	0.39	60

The ablative techniques were conducted by one skilled operator (S.S.) of a tertiary thyroid center. There were no major complications or minor complications.

## Data Availability

The data are deposited in an electronic database at the Department of Endocrine and Ultrasound-guided Surgery, “Ospedale del Mare”, Naples, Italy.

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
