# Peer review of "Laser Ablation Treatment of Recurrent Lymph Node Metastases from Papillary Thyroid Carcinoma"

_jcm, 2021, doi:10.3390/jcm10225295_

Round 1

Reviewer 1 Report

The authors have tried to describe the efficacy of laser ablation in patients with metastatic lymph nodes from previously resected papillary thyroid carcinoma. However, there are some major issues that need to be addressed:

Materials and methods:

- It would be useful to report the histological subtype of PTC.

- A 6-month follow-up is relatively short and the conclusions are not very safe. Further follow-up is required to conclude effective treatment with laser ablation.

- Tg levels during follow-up are not reported. The patients can not be considered disease free if Tg levels remain increased even if the FNAC displays absence of Tg in the ablated Ln.

Results: it is reported that the patients had a single lymph node recurrence. The presence of infiltrated LNs in the lateral compartments may be a sign that there are multiple LNs microscopically infiltrated in the central or other lateral compartments. Scintigraphic results are not reported.

Minor issues: English language check is required.

Author Response

The authors have tried to describe the efficacy of laser ablation in patients with metastatic lymph nodes from previously resected papillary thyroid carcinoma. However, there are some major issues that need to be addressed:

Thanks to Reviewer for the appreciable comments and precious suggestions.

Materials and methods:

- It would be useful to report the histological subtype of PTC.

Thanks for your suggestion. We selected patients diagnosed with classic variant papillary carcinoma. it was specified in line 158-19.

- A 6-month follow-up is relatively short and the conclusions are not very safe. Further follow-up is required to conclude effective treatment with laser ablation.

Thanks for your suggestion. We agree that 6 months of follow-up is little to say that it is a safe treatment. The data presented are preliminary data, which already show excellent results, but we are aware that such a short follow-up is a limitation of the study. We specify it in lines 255-260.

- Tg levels during follow-up are not reported. The patients can not be considered disease free if Tg levels remain increased even if the FNAC displays absence of Tg in the ablated Ln.

Thanks for your suggestion. During post-surgical and post-RAI follow-up, serum Tg levels were assessed. Relapse was considered in cases of elevated serum Tg levels and suspicious imaging. Only then was an FNC of the suspected lesion performed. During the post-LA follow-up, however, there were no cases of persistence of serum Tg, so in association with the FNC of the treated lesion, the patients were considered free from pathology. In line 186 the normal adjective has been changed to absent as it could give rise to misunderstandings.

Results: it is reported that the patients had a single lymph node recurrence. The presence of infiltrated LNs in the lateral compartments may be a sign that there are multiple LNs microscopically infiltrated in the central or other lateral compartments. Scintigraphic results are not reported.

Thanks for your suggestion. Unfortunately, the scintigraphy also shows important limits for the research of lymph node micrometastases. All patients underwent post-RAI scintigraphy. Enrolled patients underwent neck ultrasound every 3 months during the first year of post-surgical follow-up and 1-, 3-, 6-month ultrasound for the study. During the post-treatment follow-up, no further metastatic lymph nodes were identified and an increase in serum Tg values ​​was found, indicating any residual disease. A broader case series and a follow-up extended to 24 months will be published shortly.

Minor issues: English language check is required.

Thanks for your suggestion. An English language review was performed.

Reviewer 2 Report

the work presented is interesting and involves several specialists.
There are some clarifications regarding the use of FGD PT/tc in staging,
the insert use of Radioguided occult lesion localization of cervical recurrences,
and the
preferable use of such techniques in more advanced ages where surgery can have real complications.

line 143   : specify whether levothyroxine therapy is at suppressive doses

line 83 :     use of fdg pet-tc fgd for more accurate staging

line 231 : lymphadenopathy located posterior to the cervical great vessels and inaccessible to the LA

line 33 : insert use of Radioguided occult lesion localization of cervical recurrences

Author Response

the work presented is interesting and involves several specialists.
There are some clarifications regarding the use of FGD PT/tc in staging,
the insert use of Radioguided occult lesion localization of cervical recurrences,
and the preferable use of such techniques in more advanced ages where surgery can have real complications.

Thanks to Reviewer for the appreciable comments and precious suggestions.

line 143   : specify whether levothyroxine therapy is at suppressive doses.

Thanks for your suggestion. We explained in line 162.

line 83 :     use of fdg pet-tc fgd for more accurate staging.

Thanks for your suggestion. All patients underwent post-RAI scintigraphy, but we did not perform FDG PET / TC as metastatic lymph nodes were easily identified on neck ultrasound. We do not perform FDG PET / TC in patients with suspected metastatic disease as it is burdened with false positives or false negatives and in cases of lymph node micrometastases it does not allow to make a diagnosis.

line 231 : lymphadenopathy located posterior to the cervical great vessels and inaccessible to the LA

Thanks for the clarification. We screened and treated patients with accessible lymph node metastases. Patients with inaccessible lymph node metastases were referred for other treatments.

line 33 : insert use of Radioguided occult lesion localization of cervical recurrences

Thanks for your suggestion. The use of the ROLL technique is indicated in cases where metastases may be difficult to identify during surgery, but it is not a technique used in minimally invasive treatments. On lines 35-36, we have specified better what the surgical intervention of lymph node metastases consists of.

Round 2

Reviewer 1 Report

The authors have addressed all the raised issues by the reviewers and have significantly improved the scientific soundness of the manuscript.